# Circulating Tumor Cells in Hepatocellular Carcinoma: A Comprehensive Review and Critical Appraisal

**DOI:** 10.3390/ijms222313073

**Published:** 2021-12-03

**Authors:** María Lola Espejo-Cruz, Sandra González-Rubio, Javier Zamora-Olaya, Víctor Amado-Torres, Rafael Alejandre, Marina Sánchez-Frías, Rubén Ciria, Manuel De la Mata, Manuel Rodríguez-Perálvarez, Gustavo Ferrín

**Affiliations:** 1Maimónides Institute of Biomedical Research (IMIBIC), University of Córdoba, Avda. Menéndez Pidal s/n, 14004 Córdoba, Spain; lolaespejo@gmail.com (M.L.E.-C.); gonrusan@gmail.com (S.G.-R.); h22zaolj@gmail.com (J.Z.-O.); ep2amtov@uco.es (V.A.-T.); alejandrealt@gmail.com (R.A.); rubenciria@gmail.com (R.C.); mdelamatagarcia@gmail.com (M.D.l.M.); gusfesa@gmail.com (G.F.); 2Department of Hepatology and Liver Transplantation, Reina Sofía University Hospital, Avda. Menéndez Pidal s/n, 14004 Córdoba, Spain; 3Department of Pathology, Reina Sofía University Hospital, Avda. Menéndez Pidal s/n, 14004 Córdoba, Spain; marinasanchezfrias@gmail.com; 4Department of Hepatobiliary Surgery and Liver Transplantation, Reina Sofía University Hospital, 14004 Córdoba, Spain; 5Centro de Investigación Biomédica en Red de Enfermedades Hepáticas y Digestivas (CIBERehd), 28029 Madrid, Spain

**Keywords:** hepatocellular carcinoma, liquid biopsy, circulating tumor cells, dynamic changes, therapeutic target

## Abstract

Hepatocellular carcinoma (HCC) is the fifth most common neoplasm and a major cause of cancer-related death worldwide. There is no ideal biomarker allowing early diagnosis of HCC and tumor surveillance in patients receiving therapy. Liquid biopsy, and particularly circulating tumor cells (CTCs), have emerged as a useful tool for diagnosis and monitoring therapeutic responses in different tumors. In the present manuscript, we evaluate the current evidence supporting the quantitative and qualitative assessment of CTCs as potential biomarkers of HCC, as well as technical aspects related to isolation, identification, and classification of CTCs. Although the dynamic assessment of CTCs in patients with HCC may aid the decision-making process, there are still many uncertainties and technical caveats to be solved before this methodology has a true impact on clinical practice guidelines. More studies are needed to identify the optimal combination of surface markers, to increase the efficiency of ex-vivo expansion of CTCs, or even to target CTCs as a potential therapeutic strategy to prevent HCC recurrence after surgery or to hamper tumor progression and extrahepatic spreading.

## 1. Introduction

Hepatocellular carcinoma (HCC) is the most prevalent type of liver cancer, representing 90% of primary liver malignancies in patients with cirrhosis. According to the World Health Organization, HCC is the fifth most common neoplasm worldwide and the third leading cause of cancer-related death [1,2]. The geographic distribution of HCC is very heterogeneous and closely associated with the prevalence of the different etiologies of chronic liver disease [3,4,5]. In recent decades, different healthcare strategies such as vaccination campaigns, development, and liberal prescription of direct-acting antivirals against hepatitis C, and promotion of healthy lifestyle, have been implemented to reduce the incidence of HCC [5,6], but liver cirrhosis is still prevalent and forms a major driver of HCC [7]. The Barcelona Clinic Liver Cancer (BCLC) classification is nowadays the most widely used staging system of HCC [8,9], which also provides therapeutic guidance, and it has been recommended by international clinical practice guidelines [5,10]. Patients with early-stage HCC (BCLC 0-A) may be candidates for potentially curative therapies such as radiofrequency ablation, surgical liver resection (LR), or liver transplantation (LT), with 5-year survival rates of 50–80%. On the other hand, patients with advanced HCC (BCLC C) are candidates for systemic therapies, mainly (multi) tyrosine kinase inhibitors such as sorafenib or regorafenib, and the more recently approved combination of atezolizumab (anti-programmed death-ligand 1 -PD-L1- monoclonal antibody) and bevacizumab (anti-vascular endothelial growth factor monoclonal antibody), which may provide median survival rates shorter than 2 years [4].

Therefore, early diagnosis of HCC is paramount in patients with liver cirrhosis to maximize survival [11,12]. Nowadays, patients with advanced liver fibrosis or cirrhosis may undergo screening for HCC, which consists of an abdominal ultrasound every six months. Serum biomarkers of HCC such as alpha-fetoprotein (AFP), AFP-L3 (a glycoform of AFP), des-gamma-carboxyprothrombin, glypican-3 (GPC3), or osteopontin [13] may provide information about the biological aggressiveness of HCC but, unfortunately, they are not sufficiently accurate to form part of screening strategies. There is a need for new biomarkers of HCC to refine screening and diagnosis, to better predict tumor recurrence after surgical or ablative therapies, and to monitor response in patients receiving systemic therapies. In the present review, we have comprehensively evaluated the current evidence supporting the quantitative and qualitative assessment of circulating tumor cells (CTCs) as potential biomarkers of HCC.

## 2. Liquid Biopsy and CTCs

The term liquid biopsy refers to the detection of cancer byproducts mainly in the bloodstream which could provide first-hand information about the features of the primary tumor from which they belong. Liquid biopsy offers promising benefits in terms of refined diagnosis, prognostic stratification, and therapeutic guidance in different types of cancers such as metastatic breast, colorectal, or prostate cancer [13]. There are several modalities of liquid biopsy, including the detection of CTCs, extracellular vesicles, or circulating tumor nucleic acids (ribonucleic acid -RNA-, microRNA -miRNA-, tumor deoxyribonucleic acid -DNA-). The most evident advantage of liquid biopsy is the broadly accessible study specimens, which usually consist of peripheral blood samples [14]. CTCs were first described in 1869 as malignant cells derived from the primary or metastatic tumor, able to access the systemic circulation [15,16]. They are very scarce in peripheral blood, with a very short circulating half-life (1 h to 2.4 h) [17]. Despite this, CTCs can be detected even at earlier tumor stages, being associated with a more aggressive biological behavior. Unlike other liquid biopsy markers such as cell-free DNA, CTCs are unequivocally associated with the presence of viable tumors (even if not detected by conventional imaging techniques) and the progressive increase of CTCs indicates tumor spreading recurrence and metastases.

Prior to the identification of CTCs, blood samples require an enrichment method to purify such a minority cell population. Methods for CTC enrichment may be classified into positive and negative. These are not mutually exclusive and can therefore be combined to optimize the capture of CTCs. Negative-enrichment methods are based on the removal of white blood cells by antibody-based depletion strategies usually based on anti-CD45-coated magnetic beads or CD45 depletion cocktails. Enriched CTCs can be detected afterward by several different methodologies such as immunocytochemistry (ICC) [18,19,20,21,22,23], quantitative real-time polymerase chain reaction (PCR) [24,25,26], flow cytometry, or immunofluorescence in situ hybridization (iFISH) [27,28,29,30,31,32], which may combine several probes or antibodies to identify different CTC subpopulations. In contrast, positive-enrichment methods take advantage of specific biological and physical properties of CTCs to distinguish them from non-tumoral blood cells. Physical enrichment strategies are frequently based on centrifugation and microfiltration [33,34,35,36,37,38,39,40,41,42,43,44,45,46,47,48,49,50] to selectively isolate CTCs according to their differential density and size, respectively [51]. Among all the positive-enrichment methods, immunoaffinity is the most widely used but the optimal biomarker matching all CTC subpopulations is unknown. Different combinations of antibodies can be immobilized on the surface of magnetic particles or microdevices with increased capture efficiency [51,52]. As with negative-enrichment methods, isolated CTCs can be subsequently identified by common methodologies (Figure 1). The sensitivity for the detection of CTCs varies greatly between the different methods used, and even between different studies that use the same methodology.

The only CTC detection system approved by the Food-and-Drug Administration is the CellSearch^®^ system (Menarini-Silicon Biosystems, Bologna, Italy), which is based on immunocapture (positive enrichment) of CTCs expressing epithelial-to-cell adhesion molecule (EpCAM^+^ CTCs). Although the CellSearch^®^ system has been widely used in HCC research with promising results [26,53,54,55,56,57,58,59,60,61,62], novel methodologies are in the pipeline claiming for increased CTC detection rate, sensitivity, and specificity [30,35,63,64]. Thus, CTC analysis by the IsoFlux^®^ system [64] or by negative enrichment and iFISH [30] showed an increased sensitivity in HCC candidates for LT compared to the CellSearch^®^ system (90.5% vs. 4.7% and 70% vs. 26.67%, respectively).

## 3. Surface Markers to Identify CTCs in HCC and Clinical Significance

The key to ensuring adequate sensitivity and specificity is to identify a surface marker to be targeted for optimal positive cell identification. Recent advances have shown that there is no single optimal marker, and probably the best approach is to combine different surface markers with additional selection filters according to physical properties. In this section, we provide an insight into the most widely used CTC markers in HCC with their benefits and limitations.

### 3.1. Liver and HCC Specific Markers

A logical source of candidate markers for the detection of CTCs in HCC is the liver-specific or HCC-specific proteome. Since hepatocytes do not circulate under physiological conditions, any cell expressing a liver or HCC-specific marker detected in the bloodstream may be considered a potential CTC. GPC3 is a cell membrane-anchored protein currently used in clinical practice for HCC pathological analysis and characterization [65]. Since GPC3 is more frequently observed in moderately and poorly differentiated HCC tumor cells, which are more prone for extrahepatic spreading, the presence of GPC3^+^ CTCs may inform about a de-differentiated metastatic HCC. Thus, the immunomagnetic enrichment of GPC3^+^ CTCs and subsequent analysis by fluorescent activated cell sorter with anti-cytokeratin (CK, an epithelial marker) antibodies was postulated as a valid methodology to identify poor prognostic HCC patients. Indeed, an increased number of CTCs obtained by this approach was associated with microvascular invasion (a well-known poor prognostic histological feature [66]), increased tumor recurrence rates, and shorter overall survival [67]. Asialoglycoprotein receptor (ASGPR) is a transmembrane protein exclusively expressed on the surface of hepatocytes [68], but unlike GPC3, it is highly expressed in well-differentiated HCC. Xu et al. developed a system based on magnetic beads to capture ASGPR^+^ CTCs, which were subsequently identified by ICC with antibodies anti-hepatocyte-specific hepatocyte paraffin 1 (HepPar 1; a marker for differential diagnosis of HCC [69]) or anti-CK alone. The positivity rate and the number of CTCs isolated by this technique were associated with increased tumor burden, macrovascular invasion, and poor histological differentiation [70]. This methodology was modified afterward using a new anti-ASGPR monoclonal antibody for the isolation of CTCs and a combination of anti-CK and anti-CPS1 (an antigen for HepPar 1) antibodies, thus resulting in higher sensitivity for CTC detection (89% vs. 81%) [71]. The authors designed a platform to capture and release living HCC CTCs for their ulterior expansion in a 3D cell culture assay [72]. The combination of EpCAM, ASGPR, and GPC3 antibodies in the so-called NanoVelcro CTC Assay resulted in HCC CTCs detection in up to 97.6% of patients with HCC at various stages, being the number of CTCs associated with clinical outcomes. In addition, the NanoVelcro CTC Assay identified a vimentin-positive subpopulation of CTCs that indicated a particularly aggressive biological tumor behavior [52]. Using a labyrinth device that exploits the physical characteristics of CTCs and a combination of HCC-specific antibodies against GPC3, glutamine synthetase, and HepPar-1 for ICC analysis, Wan et al. detected CTCs in 88.1% of patients with HCC and correlated the positivity rate of CTCs with the tumor stage [73]. Liver-specific or HCC-specific markers could also be useful for the identification of HCC CTCs after removal of CD45^+^ cells and/or red blood cells. Immunofluorescence analysis of negative-enriched CTCs from HCC patients with antibodies against ASGPR and CPS1 or GPC3 showed high sensitivity and specificity for detecting CTCs and could provide relevant prognostic information [18,74]. Thus, the incorporation of several specific markers into a single platform for the isolation and/or detection of HCC CTCs across all tumor stages may be more efficient than a single marker.

### 3.2. Epithelial Markers

EpCAM is one of the most used membrane-associated proteins for capturing CTCs in peripheral blood samples as it is not expressed by normal blood cells [75]. EpCAM^+^ CTCs have been associated with different clinicopathological features of HCC, including vascular invasion, high serum AFP (≥400 ng/mL), and more advanced BCLC stage [58,59]. In patients undergoing LT, LR, transarterial chemoembolization (TACE), or radiotherapy for HCC, EpCAM^+^ CTC count at baseline has been associated with increased tumor recurrence, shorter overall survival, and shorter progression-free survival [26,54,55,60]. EpCAM^+^ CTCs could also aid to assess the optimal surgical margin size during LR [62], optimizing the management of HCC patients on the waiting list for LT [76], and for a better selection of candidates to certain therapies [61]. In other words, EpCAM^+^ CTCs are strongly correlated with tumor aggressiveness and could assist in the adequate stratification of HCC patients for optimal therapy and in predicting their response to treatment [56]. Unfortunately, EpCAM-based CTC enrichment may overlook CTCs with low expression of the epithelial marker [77,78] or those which expression has been lost during the dynamic process of epithelial-to-mesenchymal transition (EMT) [75,79]. Thus, the use of additional criteria such as size and brightfield image could enhance CTC detection sensitivity in HCC [22]. In addition, since EpCAM has been identified as a surface marker of HCC cells displaying stem cell features [80,81] and EpCAM^+^ CTCs from HCC patients are highly tumorigenic in vivo [60], it is possible that EpCAM^+^ CTCs account for both tumorigenic and non-tumorigenic cells.

### 3.3. Epithelial-to-Mesenchymal Transition Markers

The EMT is the process by which a CTC with an epithelial phenotype acquires mesenchymal characteristics that confer increased migratory ability, invasiveness and/or drug resistance, thus contributing to HCC spreading [82]. The activation of the EMT process in CTCs occurs primarily in the bloodstream and involves dynamic adaptive mechanisms that are associated with stress response, cell cycle, or immune evasion [21,40,83,84]. This process may not be considered dichotomous: CTCs do not necessarily exist in ‘pure’ epithelial or mesenchymal states. CTCs can be found in intermediate states showing characteristics of both epithelial and mesenchymal phenotypes [85]. Therefore, the use of a combination of epithelial and mesenchymal markers would improve the sensitivity and the overall accuracy of a CTC enrichment method. The expression of EMT markers such as vimentin, twist, Zinc finger E-box-binding (ZEB)1, ZEB2, snail, slug, and E-cadherin has been studied in liver-derived CTCs from HCC patients, being the co-expression of twist and vimentin significantly correlated with tumor burden, macrovascular invasion, and more advanced stages [86]. The CanPatrol^TM^ system, which combines a positive-enrichment filter-based method and an RNA-in situ hybridization (RNA-ISH) technique, was able to stratify CTCs into three phenotypic subgroups accordingly to the predominantly expressed markers: epithelial phenotype (EpCAM, CK 8/18/19), mesenchymal phenotype (vimentin, twist), and mixed/hybrid phenotype. Overall CTC count (which comprises all phenotypes) performed better than serum AFP to discriminate between patients diagnosed with HCC and those with non-malignant liver diseases [44]. Mesenchymal CTCs (mCTC) and mixed CTCs were associated with clinical features indicating poor prognoses, such as high serum AFP levels, advanced tumor stages, and microvascular invasion. Thus, EMT-based classification of CTCs could predict early recurrence of HCC, metastases, and shorter overall survival [33,36,40,43,45,48,50].

Epithelial CTCs and mCTC are occasionally found in the bloodstream bound to immune cells, platelets, and fibroblast as circulating tumor microemboli or CTC clusters, which may arise from the detachment of multicellular aggregates from the primary tumor mass or the reaggregation of CTCs in the bloodstream [84]. These clusters contribute to the survival and adoption of invasive advantages of CTCs and highlight the relevance of the tumor environment in the metastatic process [87,88]. The microfluidic isolation and subsequent immunofluorescence identification of CTC clusters in HCC patients with more advanced tumor stages indicated their association with tumor progression [73]. By using the CanPatrol^TM^ system, Luo et al. showed that the presence of CTC clusters consisting of CTCs (epithelial and/or mesenchymal) and white blood cells was associated with tumor burden and vascular invasion, being an independent predictor of poor prognosis in HCC [41]. In addition, the authors correlated the number of mCTC and CTC clusters with a specific mutational profile of EMT-related genes involved in the metastatic process of HCC [42].

### 3.4. Stem-Cells Markers

Stem cell markers are expressed in a small subpopulation of CTCs which may be considered tumor-initiating cells related to increased aggressiveness and poor prognosis. These circulating cancer stem cells (cCSC) may derive directly from the tumor or an adaptive process during the EMT. Due to its scarcity in peripheral blood, CTC-derived xenograft models and CTC-derived ex vivo cultures have been used for the study of cCSC [89]. CSCs may express surface markers of regular stem cells such as EpCAM, CD133, CD44, CD90, or ICAM-1. CD44 is an adhesion molecule that facilitates tumor cell invasion and migration [90] and it is concomitantly expressed with other stem-cell markers. In HCC cells and tumor xenograft models, CD133^+^CD44^+^ cells (but not CD133^+^CD44^-^) display stem cell properties, including extensive proliferation, self-renewal capacity, tumorigenic potential, resistance to chemotherapy, and expression of stem cell-associated genes [91]. In this sense, CD133^+^CD44^+^ cCSC has been correlated with increased serum transaminases, serum AFP and poorer outcomes in HCC patients [92]. CD90^+^CD44^+^ cells represent a more aggressive cell subpopulation within the CD90^+^ group, with increased in vivo tumorigenic capacity than CD90^+^CD44^-^ cells. Interestingly, CD90^+^ CTCs express higher levels of CD44 than those cells located within the primary tumor, suggesting increased tumor aggressiveness. Indeed, the number of CD90^+^ cCSC was related to disease progression in HCC patients [93]. The concomitant expression of CD90 with other surface markers such as CXCR4 (receptor for stromal cell-derived factor-1 (SDF-1/CXCL12)) may be required by cCSC to motivate HCC progression [94]. EpCAM^+^ CTCs from HCC patients exhibit high tumorigenic activity in vivo and were associated with the expression of others CSC markers such as CD133 and ABCG2 [60]. Based on a strategy combining negative cell enrichment and quantitative real-time PCR, the expression of nine CSC markers was analyzed in the blood of HCC patients. Aligning with the above-referred studies, EpCAM, CD133, CD90, and CK19 (another hepatic stem marker [95]) were significantly overexpressed in the HCC group and associated with the presence of a corresponding subpopulation of HCC CTCs and with a higher tumor recurrence rate [25]. Similarly, EpCAM^+^CD90^+^ CTCs have been associated with HCC recurrence after LT [96]. Although expression levels of ICAM-1 did not differ significantly from the control group in that study, hepatic ICAM-1^+^ CTCs from HCC patients show high tumorigenic activity in vivo and their frequency is an independent risk factor of portal vein tumor thrombus and ascites [97].

## 4. Dynamic Changes of CTC Counts after HCC Therapy

Preoperative CTCs have been widely associated with HCC recurrence after LR [24,36,55,56,60], LT [29,30,96] or local ablation [28]. Theoretically, a complete tumor removal by LR or LT may motivate a rapid decline in CTC count in the post-operative period. Similarly, local ablation therapies could release CTCs to the bloodstream initially, followed by a progressive decline until complete clearance. In patients receiving systemic therapies, the dynamic of CTC counts may vary depending on the therapeutic response. In any of these situations, trends of CTC count after therapy could provide valuable information to predict tumor recurrence or to anticipate resistance to systemic therapies (Figure 2).

### 4.1. CTC Dynamics after Liver Resection

Studies evaluating intraoperative CTC monitoring have shown that surgical manipulation during LR is not associated with a significant CTC release into the bloodstream [49]. LR appeared to have little effect on CTC count in the immediate postoperative period [53]. The decline of CTC count after surgical resection may be more evident between postoperative days 7 to 10 [40,49] and could continue declining up to the first post-operative month [26,37,56,60]. The postoperative increase in CTCs and the persistence of a high CTC count after LR may increase the risk of tumor recurrence and extrahepatic metastases, as well as shorter overall survival. These results were consistent using the CellSearch^®^ [26,53,56,60] and the CanPatrol^TM^ [37,40] systems, as well as the isolation by size of epithelial tumor cells method [49]. It may well be that these patients with persistence of CTCs had an incomplete tumor removal or additional HCC microscopic foci within the liver, which in the last term would motivate local recurrence or distant metastases. The percentage of mCTC (and total CTC count) could anticipate tumor recurrence or metastases up to two months before it is evident in imaging techniques [40].

### 4.2. CTC Dynamics after Liver Transplantation

LT is considered a more radical therapy than LR for patients with HCC since the whole cirrhotic liver is removed, including the known HCC nodules and the possible nascent tumors in the remaining liver parenchyma. A more rapid decline of CTCs would therefore be expected. A single study using negative-enrichment and iFISH platform showed a significant decrease in CTC count 3 months after surgery in most patients [30]. However, CTC dynamics assessed by the CanPatrol^TM^ system did not find a significant association with HCC recurrence [34]. More recently, Wang et al. reported that those patients who did not have CTCs pre-LT and showed positive counts 2–4 weeks after LT had an increased risk of tumor recurrence [20]. This study used a negative-enrichment method and ICC detection of CTCs and, unlike the results obtained by Xue et al. by using iFISH [30], no significant difference in preoperative CTC count or CTC-positive rate was observed between patients who experienced tumor recurrence and those who did not. Very recently, Hwang et al. have shown that CTCs with cancer stemness could predict HCC recurrence when they are detected preoperatively and 1 day after LT using fluorescent activated cell sorter [96]. We have evaluated the clearance kinetics of CTCs in HCC patients undergoing LT and found that the presence of CTC clusters before surgery was associated with an incomplete clearance of CTCs at postoperative day 30, which in turn predicted mortality due to extrahepatic recurrence of HCC [98]. The unpredictable evolution of CTCs after LT could be in part explained using immunosuppressive drugs, which are required to prevent graft rejection. Since evading the immune system is one of the hallmarks of cancer [99], the use of immunosuppressive agents such as calcineurin inhibitors may increase the risk of tumor recurrence in a dose-dependent manner [100]. A reduced CTC count after LT in a patient deeply immunosuppressed may be sufficient to persist in the bloodstream leading to tumor recurrence. Different protocols including CTCs monitoring have been proposed to decrease the risk of tumor recurrence after LT including the following aspects: (1) an adequate selection of recipients with limited tumor burden and reduced baseline CTC count, (2) minimizing the handling of large HCCs during transplantation to decrease the risk of CTC release, (3) decreasing liver graft ischemia-reperfusion injury to prevent the engraftment of CTCs in the liver and (4) consider using anticancer drugs and (5) minimization of immunosuppression [101].

### 4.3. CTC Dynamics after Local Ablative Therapies and TACE

It has been suggested that radiofrequency ablation [46] and TACE [31,102] could motivate the release of CTCs. Conversely, microwave ablation has been associated with a significant decrease of CTCs and a higher tumoricidal effect [103]. Dynamic changes of CTCs after radiofrequency ablation [39], chemoembolization [31,103,104], or microwave ablation [39,103] seemed to have no impact on tumor recurrence rates when CTCs were identified by using the CanPatrol^TM^ system, negative-enrichment, and flow cytometry or iFISH, and immunomagnetic enrichment. However, these studies were limited by the reduced sample size. In a study including 155 HCC patients assessing CTCs by negative-enrichment and iFISH, patients who responded to TACE (complete response and partial response) had a significant CTC count decline whereas patients without radiological response did not [32]. Similarly, when CTCs were assessed by the CellSearch^®^ system, patients with stable or decreasing CTC count one month after TACE or radiotherapy showed radiological response or stable disease, while most of the patients with persistently high CTC count showed disease progression [26].

### 4.4. CTC Dynamics during Systemic Treatment

There is no evidence regarding the dynamics of CTCs in patients with HCC receiving multikinase inhibitors or immunotherapy. In a highly metastatic orthotopic nude mouse model of green fluorescent protein-labeled HCC, Yan et al. used a non-invasive in vivo flow cytometry method and showed that sorafenib could reduce the number of CTCs by inhibiting tumor proliferation and angiogenesis [105]. A study evaluating cell-free DNA has identified a mutational profile that indicates resistance to sorafenib [106]. More reliable information could be obtained by analyzing the dynamic of CTC counts and by characterizing the mutational profile of CTCs longitudinally to anticipate therapeutic resistance and radiological progression, thus allowing patients to access a second-line therapy in a timely manner.

## 5. Cell Culture of HCC CTCs

Ex vivo expansion of viable CTCs can be useful to obtain prognostic information and to investigate the biochemical, gene expressional, and behavioral properties of the primary cancer cells [107]. CTC culture would allow identifying specific targets for cancer therapy, monitoring the acquisition of mutations conferring drug resistance, and even testing the vulnerability of the tumor against multiple possible therapeutic agents in a non-invasive manner [108]. Compared to preclinical CTC-derived xenograft models, the ex vivo culture of CTCs is less time-consuming, which can be paramount when making individualized clinical decisions [89]. Researchers frequently use 3D culture conditions and the high glucose DMEM/F12 medium with epidermal growth factor and/or basic fibroblast growth factor for the enrichment of cancer cell lines-derived CSC spheroids in vitro [109,110]. Optional supplements are N2, B27, inhibitors Y27632 and SB431542, heparin, insulin-transferrin-selenium, or fetal bovine serum. Even though, CTC culture is challenging. Stable ex vivo culture of cCSC has been successfully achieved mainly in patients with metastatic cancer and increased CTC count [111,112], but CSC spheroids often survive for only a few days or weeks [113]. Therefore, ex vivo CTC expansion is inefficient even in patients with advanced cancer [114,115].

Ex vivo culture of cCSC in patients with HCC has been reported even more anecdotally. The few successful studies included intermediate/advanced cancer patients and reported similar culture conditions including normoxia, Matrigel^TM^ solution, and DMEM (with or without 10% fetal bovine serum). cCSC isolation methods varied widely, including negative selection with magnetic beads [23,116] and positive selection by microfluidic chip [72] or fluorescent activated cell sorter [117]. A success rate of 86% to 100% has been reported with a median cell survival time of 7–14 days, although a publication bias is highly probable.

The main caveat for CTC culture is again the reduced number of these cells in the bloodstream. Different approaches are under investigation to increase the isolation rate of CTCs [118]. New methodologies for stable ex vivo culture of CTCs will need to mirror the tumor microenvironment as accurately as possible. Stress conditions naturally occurring in solid tumors such as glucose depletion and hypoxia have been tested. Glucose deprivation slows down cell proliferation but increases the expression of stem cell markers and other stem cell phenotypic characteristics [119]. Regarding the oxygenation conditions, the culture of cCSC has been successfully achieved both in normoxia and hypoxia. Hypoxia is a critical microenvironmental factor promoting the self-renewal of CSCs and a more aggressive phenotype in tumor cells [83,110,111,120], which in turn is associated with resistance to therapy [121,122]. Hypoxic conditions have been successfully used for long-term CTC culture from patients with different types of cancer including colorectal [123], lung [124], breast [89,108], or gastroesophageal [125] tumors. In an ongoing research project analyzing the clearance of CTCs after local ablation of HCC, we tried to obtain cCSC spheroids by using different strategies. First, we used anti-EpCAM-coated immunomagnetic beads and the IsoFlux^®^ system (Fluxion Biosciences, Alameda, CA, USA), which displays great sensitivity in the identification of HCC-derived CTCs [64]. Although we were able to obtain CSC spheres from established HCC cell lines using this strategy, we failed to obtain cCSC spheroids in any of the 14 patients analyzed after 2 weeks of cell culture in normoxia, high glucose, and ultra-low adherence. By contrast, in an independent cohort of 12 patients, the negative enrichment and subsequent culture of CD45^-^ CTCs under the same conditions allowed to obtain CSC-like spheroids in 50% of patients. Despite this encouraging result, spheroids could not be expanded for more than 7–10 days and they finally died off after this period (data not shown). In addition to the difficulties of cultivating CTCs, it is worth mentioning that most patients included in this study were intermediate-stage HCC patients, with lower CTC counts than those obtained in advanced HCC.

## 6. CTCs and Microenvironment as Therapeutic Targets in HCC

Targeting CTCs is an attractive strategy to hamper tumor progression. Indeed, the development of tumor cell populations with metastatic potential is a critical hallmark of cancer. There are several therapeutic strategies directed against the tumor microenvironment to avoid CTC survival and differentiation to more aggressive phenotypes, which are shown in Table 1.

### 6.1. CTCs as Therapeutic Targets in HCC

The approved multikinase inhibitors for advanced HCC may affect the viability of CTCs. Adjuvant therapy with sorafenib after surgical resection could eliminate residual CTCs to reduce the risk of postoperative recurrence and metastases [145]. pERK^+^/pAkt^−^ CTCs are vulnerable to sorafenib and their clearance is associated with improved progression-free survival [23]. However, the most effective therapy against CTCs would be their physical removal before they can migrate to other organs and produce metastases. The Viatar cancer dialysis system (Viatar CTC Solutions Inc., Lowell, MA, USA), the temporary indwelling intravascular CTC isolation system [126], the GILUPI CellCollector^®^ [127] or the black phosphorus and antibody-functionalized intravenous catheter [128] are different technologies designed for in vivo removal and destruction of EpCAM^+^ CTCs [128]. Most of them were designed to allow the screening of a large volume of blood and they could be used as adjuvant therapies after surgical resection or locoregional ablation. Although each of these systems claims for inherent technical advantages, additional studies are needed to determine their safety and efficacy in vivo.

Gene therapy and immunotherapy have also been proposed as promising tools to selectively target resistant subpopulations of CTCs. Immune checkpoint inhibitors in combination with epidermal growth factor receptor (EGFR) inhibitors (atezolizumab/bevacizumab) have recently become first-line therapies in patients with advanced HCC, as they have demonstrated improved progression-free and overall survival rates as compared with sorafenib, the previous standard of care [146]. Anti-programmed cell death protein 1 (PD-1) therapy could target PD-L1^+^ CTCs, which are associated with more aggressive disease [129]. Inhibition of EGFR signaling reduces total CTC count and metastases after local ablation [130,147]. There are other potential therapeutic targets able to interfere with the viability of CTCs which could reshape future therapeutic algorithms in HCC. Major vault protein (MVP) depletion by siRNA technology or by anti-MVP antibodies reduces survival, migratory capacity, and invasiveness of HCC cells [19]. Other strategies to target CTCs, either as a whole population or as subpopulations with particularly aggressive phenotypes are depletion of ubiquitin-specific protease 1 [133], inhibition of Intercellular Adhesion Molecule 1 (ICAM-1) using a plasmid expressing a short hairpin RNA [97], blockade of CD44 by specific antibodies [93,131] and promotion of the expression of androgen receptor, which suppresses CD90 expression and cell migration and increases cell death [132].

Interfering with the EMT activation or the colonization of CTCs could be therapeutically attractive to prevent the extrahepatic spread of HCC [84,134]. Tacrolimus, an immunosuppressant commonly used to prevent allograft rejection after LT, has been associated with the induction of motility and invasiveness of HCC cells in vitro and it is able to increase the risk of tumor recurrence in a dose-dependent manner [100]. One of the main mechanisms underlying this clinical observation is the activation of the Rho/ROCK signaling pathway. The administration of Y-27632, a ROCK-specific inhibitor, prevents tumor recurrence after LT in an animal model of HCC [135,148]. However, this mechanism should be further investigated since ROCK inhibition also promotes survival and expansion of tumor stem cells in vitro [149].

### 6.2. Tumor Microenvironment as a Therapeutic Target in HCC

Many interactions or communication factors are forming the pre-metastatic niche, which promotes the arrival, survival, and growth of tumor cells. Targeting this crosstalk could prevent extrahepatic spreading of HCC. The SDF-1/CXCR4 axis participates in the acquisition of stemness and migratory potential of HCC cells and could be considered a potential target to prevent metastases [136], with promising results obtained in preclinical studies targeting CXCR4 [137,138] or SDF-1 [139] by specific inhibitors or gene expressing traps respectively. MiRNAs can be released by many cell types into the extracellular space where they regulate different processes. Thus, the tumor-derived exosomal miR-1247-3p transforms fibroblasts into cancer-associated fibroblasts to promote lung metastases of liver cancer [140]. MiR-155 released in exosomes by HCC cells stimulates cell colony formation, migration, and invasion in vitro [141] and promotes HCC proliferation in vivo [150]. Similarly, SMAD Family Member 3-enriched exosomes derived from primary tumors promote the metastatic potential of CTCs [142].

Other interactions between CTCs and the tumor microenvironment involve the immune system [151,152,153], and particularly the adhesive myeloid-derived suppressor cells, tumor-associated macrophages, and regulatory T cells (Tregs). These interactions protect CTCs from immune responses, thus increasing their metastatic potential. Liver sinusoidal endothelial cells are the major liver cell type responsible for TGF-β-dependent hepatic Treg induction and immune tolerance [154]. The immunosuppressive chemokine CCL5 recruits Tregs to the tumor microenvironment to promote immune escape [155]. This may explain the correlation between the number of Tregs and CTC count [24]. Moreover, Treg could in turn induce CCL5 expression in CTCs by secretion of TGF-β1 [21]. Targeting liver sinusoidal endothelial cells [143] or blocking the CCL5 pathway [21,155] could interfere in this feedback, thus inducing a hostile microenvironment [144] which hampers CTC survival and metastatic colonization.

## 7. Future Perspectives

Liquid biopsy, and particularly quantitative and qualitative assessment of CTCs, will probably reshape the future diagnostic and therapeutic algorithms of HCC. This valuable information may be easily obtained from low-volume peripheral blood samples, either with diagnostic, therapeutic purposes or even to stratify patients according to the biological aggressiveness of the primary tumor. However, current methodologies pose important caveats which need to be addressed before this technology is implemented in routine clinical practice. First, the optimal combination of markers for the enrichment of CTCs, able to detect only viable cells with metastatic potential, is unknown. Most studies have a limited sample size, lack external validation and their results become obsolete shortly after their publication due to the vertiginous technological development. Secondly, isolating CTCs for ulterior culture and evaluation of their mutational profile is challenging and very inefficient under current strategies. The development of optimal culture conditions to expand CTCs is paramount to evaluate their resistance to available therapies and to advance towards truly personalized medicine in HCC. Finally, CTC counting is a time-consuming process that requires highly specialized personnel and expensive technology. Blood samples should be analyzed as soon as possible after the extraction and storing samples longer than 24 h could threaten the viability of CTCs. The whole process should be automatized, and costs reduced so this technology is made available in all institutions taking care of patients with HCC. Only then, liquid biopsy could be incorporated as part of the diagnostic and therapeutic armamentarium of HCC.

## Figures and Tables

**Figure 1 ijms-22-13073-f001:**
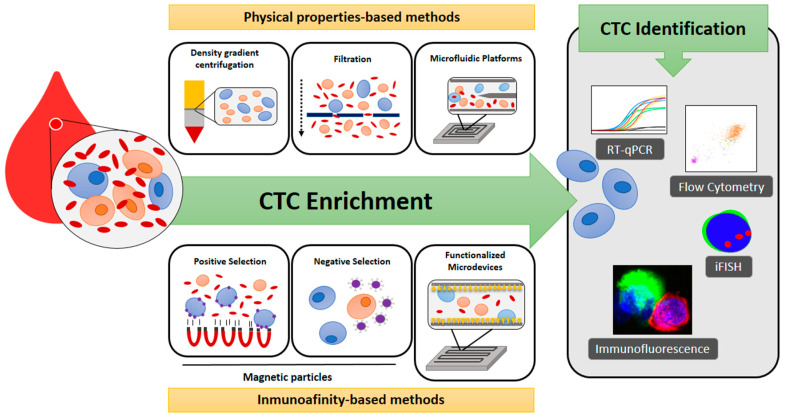
Illustration summarizing the most relevant methodologies for circulating tumor cells (CTCs) enrichment and detection.

**Figure 2 ijms-22-13073-f002:**
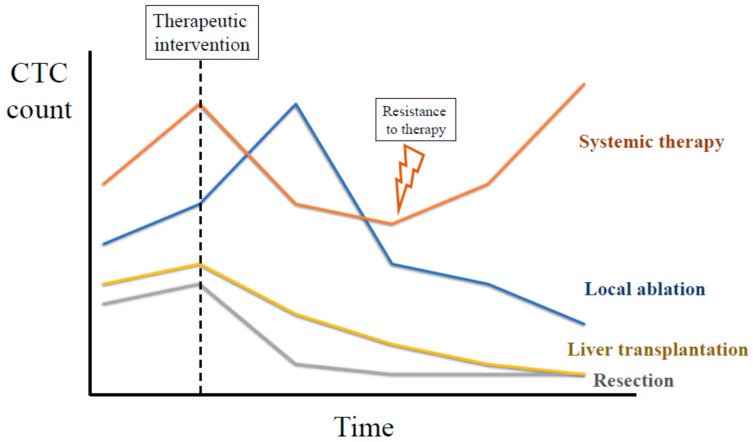
Illustration depicting the theoretical dynamics of CTCs after different successful therapeutic interventions.

**Table 1 ijms-22-13073-t001:** Potential strategies to target CTCs or tumor microenvironment to prevent HCC recurrence and metastases. The table includes systemic treatments, physical removal methods, and strategies based on gene therapy and immunotherapy.

Methodology /Therapy	Target	Study Population	Key Findings
Sorafenib	pERK^+^/pAkt^−^ CTCs	HCC patients	pERK^+^/pAkt^−^ CTCs are sensible to sorafenib [23].
Viatar System	EpCAM^+^ CTCs		Physical removing of CTCs. It requires a dialysis system. Proof of concepts.
TIIAS		In vitro, canine model	Miniaturized Viatar System. Portable aphaeretic system. It analyzes 1–2% of whole blood in 2 hours [126].
FSMW		Cancer patients	Passive CTC-capturing device. Estimated analysis of 1.5–3 liters of blood in 30 min [127].
BPAFIC		In vitro, rabbit model	Passive CTC-capturing device. Captures 2.1% of CTCs in 5 min and kills them with 100% efficiency [128].
Anti-PD-1	PD-L1^+^ CTCs	HCC patients	Favorable response to anti-PD-1 therapy is associated with the presence of PD-L1^+^ CTCs [129].
Anti-EGFR	CTCs (angiogenesis, cell migration)	Mouse model	EGFR inhibition may reduce CTCs after transarterial chemoembolization [130].
Anti-MVP	MVP^+^ CTCs	In vitro	Anti-MVP therapy target MVP^+^EpCAM^−^ CTCs, which are related to metastases [19].
Anti-CD44	CD90^+^CD44^+^ CSCs	In vitro	CD44 blockade could induce the death of CD90^+^ cells [131].
AR overexpression	CD90 expression	Mouse model	Enhancing AR expression in CTCs might reduce the risk of HCC recurrence [132].
Anti-USP1	USP1^+^ CTCs	Mouse model	USP1 upregulation in CTCs correlates with metastases and reduced survival. USP1 inhibition is a potential therapy for HCC [133].
Anti-ICAM-1	ICAM-1^+^ CTCs	Mouse model	Inhibition of ICAM-1 reduces tumor initiation and metastases [97].
Anti-TM4SF5	TM4SF5^+^ CTCs	Mouse model	Targeting TM4SF5 or interaction between TM4SF5 and CD44 may lead to efficient inhibition of TM4SF5-mediated metastases [134].
Y-27632	Rho-associated kinase	Rat model	Y-27632 inhibits tacrolimus-enhanced invasiveness of cancer cells and could be used to prevent tumor recurrence after LT [135].
CXCR4/SDF-1 axis blockade	CXCR4^+^ CTCs	In vitro, mouse model	STAT3 inhibition and CXCR4 blockade have clinical therapeutic potential in HCC [136]; Hepatic stellate cells play an important role in liver metastases by releasing SDF-1 [137,138].
	Hepatocytes	Mouse model	Non-viral SDF-1 trap gene decreases liver metastases in models of colorectal and breast cancer [139].
miR-1247-3p/IL-6, IL-8 axis blockade	miR-1247-3p/ IL-6/ IL-8	In vitro, mouse model	Tumor-derived exosomal miR-1247-3p converts fibroblasts to cancer-associated fibroblasts which promote tumor stemness, EMT, chemoresistance, and tumorigenicity [140].
miR-155 blockade	Tumor-derived miR-155	In vitro, mouse model	MiR-155 is highly elevated in EpCAM^+^HCC cells and could be an actionable target to remove the EpCAM^+^CSC population [141].
Anti-SMAD3	CTCs	In vitro, mouse model	SMAD3-containing exosomes from primary tumors could favor the viability and adhesion of CTCs and the risk of metastases [142].
Function modulation	LSECs	In vitro	Cancer-activated LSECs can enhance the proliferation of Tregs and promote cancer cell liver colonization [143].
anti-CCL5/ CCR5,CCR4	CCL5/ CCR5^+^, CCR4^+^ Tregs	In vitro, mouse model	Chemokine CCL5 recruits regulatory T cells to facilitate the immune escape of CTCs [21].
IFN-α	Microenvironment	Mouse model	IFN-α does not decrease the number of CTCs but could modulate the tissue microenvironment to resist CTCs and metastases [144].

TIIAS, Temporary Indwelling Intravascular Aphaeretic System; FSMW, Structured and Functionali-zed Medical Wire; BPAFIC, Black Phosphorus and Antibody Functionalized Intravenous Catheter; AR, Androgen receptor; CCR, colorec-tal cancer; EMT, Epithelial-Mesenchymal Transition; LSECs, Liver sinusoidal endothelial cells; Tregs, regulatory T cells.

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
