# Peer review of "Circulating Tumor Cells in Hepatocellular Carcinoma: A Comprehensive Review and Critical Appraisal"

_ijms, 2021, doi:10.3390/ijms222313073_

Round 1
Reviewer 1 Report
This paper comprehensively reviews the current status of circulating tumor cells (CTCs) in hepatocellular carcinoma (HCC). CTCs have biological information and are potentially useful for the diagnosis and therapeutics of HCC. However, the optimal methods for quantitative and qualitative CTCs assessment are still uncertain. More studies are needed for the clinical application.
Comments:
Line 72-
As "liquid biopsy" is widely meant not only using the blood but also the body fluid such as saliva, urine, pleural fluid, and so on, as a specimen, it is better to add "mainly" before "in the bloodstream-".
(Although the author mentioned later,,)
Line 75-
“many types of cancers [13]”
Please also refer to other types of cancer.
While I am not familiar with the HCC, what is the merit of detecting CTCs than circulating tumor nucleic acids in HCC?
Is there no molecular targeting agent for HCC?
How is the sensitivity of detecting circulating tumor nucleic acids or CTCs in HCC?
Line 475-
I agree CTC counting is a so time-consuming process but isn't it preserved up to almost several days by a preservative blood collection tube (solution)?
Overall, I think if there is an illustration of a methodology for CTC detection, it seems to be easy to understand.
Author Response
To the REVIEWER 1:
- Line 72-As "liquid biopsy" is widely meant not only using the blood but also the body fluid such as saliva, urine, pleural fluid, and so on, as a specimen, it is better to add "mainly" before "in the bloodstream-".(Although the author mentioned later,,)
Thank you for your comments. We have included this pertinent correction in the main text.
- Line 75-“many types of cancers [13]”. Please also refer to other types of cancer.
We have now included colorectal, prostate and breast cancer as types of cancers where liquid biopsy associated with CTC count can offer promising benefits (line 75). Thus, the CellSearch is the only system clinically validated by the FDA for the isolation and enumeration of CTCs, with predictive capacity in these patients.
- While I am not familiar with the HCC, what is the merit of detecting CTCs than circulating tumor nucleic acids in HCC?
This is a very pertinent comment. Circulating tumor cells and cell-free DNA are equally valid methodologies of liquid biopsy and there is controversy on which one will become the standard of care. In our opinion, both techniques are complementary. Cell-free DNA offers valuable information about the development of mutations conferring resistance to systemic therapies but it does not inform about the tumor burden or progression. In contrast, CTCs are associated with the presence of viable tumor (even if not detected by conventional imaging techniques) and the progressive increase of CTCs indicates tumor spreading, recurrence, and metastases. In the present review, we have focused in CTC-based methodologies of liquid biopsy since addressing both methodologies in depth is not possible in a single manuscript. We have included a sentence in the main text regarding this issue (lines 83-86).
- Is there no molecular targeting agent for HCC?
We have now included a paragraph in the introduction chapter about the antitumor agents that are currently used for the treatment of advanced HCC in order to clarify this question (lines 52-57).
- How is the sensitivity of detecting circulating tumor nucleic acids or CTCs in HCC?
This is an important issue. The sensitivity of detecting CTCs varies greatly between the different methods used, and even between different studies that use the same methodology. We have discussed this briefly in the appropriate section (section 2. Liquid biopsy and CTCs) and we have included two studies comparing the CellSearch system with more sensitive methodologies.
- Line 475-I agree CTC counting is a so time-consuming process but isn't it preserved up to almost several days by a preservative blood collection tube (solution)?
There are commercialized preservative blood collection tubes for this purpose. However, we have not used this methodology in our studies and it is also not frequently referred by other researchers for the isolation of CTCs.
- Overall, I think if there is an illustration of a methodology for CTC detection, it seems to be easy to understand.
We have included a new figure summarizing the most relevant methodologies for CTC enrichment and detection (Figure 1).
Reviewer 2 Report
Manuscript ID ijms-1462477, entitled "Circulating tumor cells in hepatocellular carcinoma: a comprehensive review and critical appraisal”.
Thank you for the opportunity to review this comprehensive review.
The study by María Lola Espejo-Cruz et al deals with a topic of interest and a question that, although not new, it is still under investigation and pertinent to our clinical practice in the field of HCC.
Liquid biopsy, and particularly circulating tumor cells have emerged as a useful tool for diagnosis and monitoring therapeutic response in different tumors.
The Authors, evaluate the current evidence supporting the quantitative and qualitative assessment of circulating tumor cells as potential biomarkers of HCC, as well as technical aspects related to isolation, identification and classification of circulating tumor cells.
Few points have the potential to be improved.
The introduction is quite long and would benefit from being streamlined and shortened.
A recently published study (doi: 10.5009/gnl210162) suggested EpCAM+/CD90+ circulating tumor cells can be used preoperatively and 1 day after LDLT as key biological markers in LT candidate selection and post-LDLT management. Please, briefly discuss the reference in the appropriate chapter.
Author Response
To the REVIEWER 2:
- The introduction is quite long and would benefit from being streamlined and shortened.
Thank you very much for your comments. The introduction has been significantly shortened in the new version of the manuscript and a brief comment was added regarding systemic therapies of HCC as suggested by the reviewer 1.
- A recently published study (doi: 10.5009/gnl210162) suggested EpCAM+/CD90+ circulating tumor cells can be used preoperatively and 1 day after LDLT as key biological markers in LT candidate selection and post-LDLT management. Please, briefly discuss the reference in the appropriate chapter.
We have included this reference in the text (ref. 96) and have discussed its content in the appropriate chapters (3.4. Stem cell markers, and 4.2. CTC dynamics after liver transplantation).